

# Exploring jet substructure in semi-visible jets

**Deepak Kar⋆ and Sukanya Sinha†**

School of Physics, University of Witwatersrand, Johannesburg, South Africa

⋆ deepak.kar@cern.ch, † sukanya.sinha@cern.ch

## Abstract

Semi-visible jets arise in strongly interacting dark sectors, where parton evolution includes dark sector emissions, resulting in jets overlapping with missing transverse momentum. The implementation of semi-visible jets is done using the Pythia Hidden valley module to duplicate the QCD sector showering. In this work, several jet substructure observables have been examined to compare semi-visible jets (signal) and light quark/gluon jets (background). These comparisons were performed using different dark hadron fractions in the semi-visible jets. The extreme scenarios where signal consists either of entirely dark hadrons or visible hadrons offers a chance to understand the effect of the specific dark shower model employed in these comparisons. We attempt to decouple the behaviour of jet-substructure observables due to inherent semi-visible jet properties, from model dependence owing to the existence of only one dark shower model as mentioned above.



# 1    Introduction

Searches for dark matter (DM) particles in colliders have remained unsuccessful so far [1]. Consequently in recent years, some focus has shifted to unusual final states, which are not covered by typical searches at the Large Hadron Collider (LHC). One such final state is termed as semi-visible jets (svj), where parton evolution includes dark sector emissions, resulting in jets interspersed with DM particles [2, 3]. While searches for semi-visible jets are underway in the LHC experiments, the focus of this paper is to probe the viability of such searches in boosted topologies.

We focus on the more challenging scenario of t-channel production mode of semi-visible jets, where the absence of a resonance mass peak makes identifying the substructure difference more critical. The two-vertex coupling strength, $\lambda$, can be treated as a free parameter to gauge the sensitivity of the semi-visible jets signal with respect to the background and the idea here is to examine if the jet substructure of semi-visible jets can be used to discriminate them from ordinary jets produced by light quarks or gluons.

We start by briefly summarising the idea of semi-visible jets in Sec. 2 and the signal cross-section sensitivity in terms of varying values of $\lambda$. Then comparisons between semi-visible jets and ordinary jets are presented in Sec. 3, based on their substructure. The robustness of these differences and the underlying reasons are investigated in Sec. 4, before concluding in Sec. 5. For these studies, the Rivet [4] analysis toolkit was used, with the Fastjet package [5] for jet clustering.

# 2    Semi-visible jets

Semi-visible jets [6] are hypothetical reconstructed collider objects where the visible components in the shower are Standard Model (SM) hadrons. It is assumed in these scenarios that the strongly coupled hidden sector contains some families of dark quarks which bind into dark hadrons at energies lower than a dark-confinement scale $\Lambda_d$. In scenarios where the mediator mass is less than the collision energy, simplistic descriptions of collider phenomenology can be constructed, keeping at bay the extra physics details of the theory, observed at energies greater than collider levels, and irrelevant at the LHC. In t-channel setups (Fig. 1), the mediator interacts with DM and one of the SM quarks, usually simplified models consider the scenario of fermionic DM particle which interacts with SM particles via a scalar mediator coupling only to right-handed quarks. A generic t-channel DM simplified model contains an extension of the SM by two additional fields: a DM candidate ($\chi$) and a mediator ($\phi$) which has it's fundamental representation in $SU(3)_c$ and the dark gauge group considered in the theory. The dark quarks are the DM degrees of freedom at LHC energies, but at scales probed by direct detection (DD) experiments, the DM degrees of freedom are dark mesons and hence DD rates are highly suppressed for such models since they fall below the neutrino background [3].

Considering the approach of parameterising the Beyond the Standard Model (BSM) effects in an EFT expansion, an effective Lagrangian captures all possible interactions:

$$\mathcal{L}_{\text{eff}} = \mathcal{L}_{\text{SM}} + (1/\Lambda)\mathcal{L}_1 + (1/\Lambda^2)\mathcal{L}_2\,,$$

where $\mathcal{L}_i$ are constructed from Standard Model operators that obey the $SU(3)_C$ x $SU(2)_L$ x $U(1)$ gauge symmetries, and the higher-dimensional Lagrangian terms representing effective (i.e. non-fundamental) couplings, are suppressed by powers of $\Lambda$. As the dark matter particles can appear in the final state of this model, a Dark Matter Effective Field Theory (DMEFT) approach is used, in which the dark matter (DM) is the only additional degree of freedom

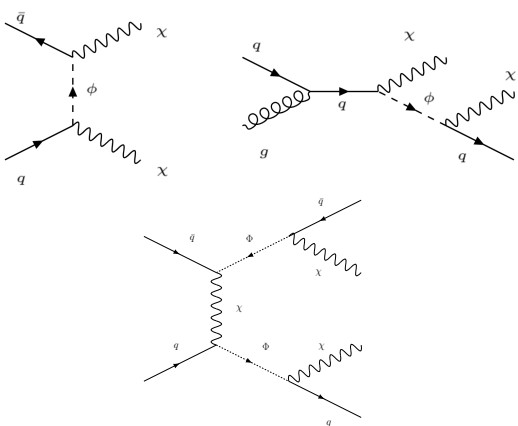

Figure 1: DM production via t-channel

beyond the SM accessible by current experiments, and hence the interactions of the DM particle with SM particles are described by effective operators (of dimension-6 or higher) of the form:

$$\mathcal{L}_{\text{contact}} \supset (c_{ijab}/\Lambda^2)(\overline{q}_i \gamma^\mu q_j)(\overline{\chi}_a \gamma_\mu \chi_b)$$

and the interaction Lagrangian is:

$$\mathcal{L}_{\text{dark}} \supset -\frac{1}{2} \text{tr}\, G^d_{\mu\nu} G^{d\mu\nu} - \bar{\chi}_a \left( i\slashed{D} - M_{d,a} \right) \chi_a \,.$$

This is a canonical Lagrangian for fermions with covariant derivatives. Here, the dark sector is a $SU(2)_D$ gauge theory with coupling $\alpha_d = g_d^2/4\pi$, containing two fermionic states $\chi_a = \chi_{1,2}$, and $c_{ijab}$ are O(1) couplings that encode the possible flavour structures, and (assuming minimal flavour-violation) heavy-flavour production channels dominate. The fermions act as dark quarks which interact strongly with coupling strength $\alpha_d$, similar to QCD. If the mediator is assumed to be unstable, they will decay to a particle which is charged under a new strong group, but is a singlet under all the SM groups. At the confinement scale $\Lambda_d$ when $\alpha_d$ becomes non-perturbative, these dark quarks form bound states [3, 6].

The spectrum of these dark hadronic states have non-perturbative dependencies, and most of the details concerning the spectrum are inconsequential for collider observables. If dark mesons exist, their evolution and hadronization procedure are currently little constrained. They could decay promptly and result in a very SM QCD like jet structure, even though the original decaying particles are dark sector ones; they could behave as semi-visible jets; or they could behave as completely detector-stable hadrons, in which case the final state is just the missing transverse momentum. Apart from the last case, which is more like a conventional BSM signature with large values of missing transverse momentum, the modelling of these scenarios is an unexplored area.

In this paper, we consider the case where the final state consists of a jet interspersed with missing transverse momentum (usually referred to as MET) due to a mixture of stable, invisible dark hadrons (with decay time $c\tau > 10$ mm) and visible hadrons from the unstable subset of dark hadrons that promptly decay back to SM particles. The model discussed in [3] uses a simplified parameterisation, where a direct mapping of the Lagrangian parameters to physical observables is not possible since some of the dark sector observables depend on non-perturbative physics. The three parameters of this model are:

- Mass of the scalar bi-fundamental mediator, $\phi$

- Dark hadron mass, $m_{\text{d}}$

- Ratio of the rate of stable dark hadrons over the total rate of hadrons, $R_{\mathrm{inv}}$

The third parameter in its intermediate regime leads to the appearance of semi-visible jets (Fig. 2). There is a possibility of production modes arising for the t-channel scenario as discussed in detail in [3], involving

- direct pair-production of the dark quarks ($pp \to \chi\bar{\chi}$),

- pair production of the bi-fundamental mediator ($pp \to \Phi\Phi^{\star}$), with $\Phi \to \chi j$

- associated production of the bi-fundamental mediator ($pp \to \Phi\bar{\chi}$), with $\Phi \to \chi j$

Figure 1 shows the representative leading order Feynman diagrams for the production of dark matter particle pairs (left), associated production of a mediator and a dark matter particle (right), and pair production of two mediators (bottom).The process generation setup for this study have been discussed in 3.1.

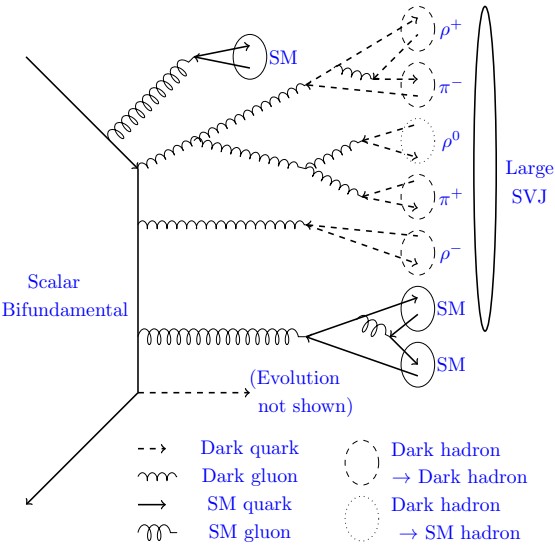

Figure 2: A schematic illustration of how semi-visible jets are produced in t-channel. Inspired by [7].

The modelling of such unique final state signatures is done using the Hidden Valley (HV) [8] module of Pythia8 [9], which was designed in order to study a sector which is decoupled from the Standard Model (SM). The basic motivation was the simple idea that one could start with a large number of gauge groups in the high energy limit, but can break them down to fewer groups as the energy decreases, while maintaining the observed cosmological bound. The module tends to achieve a reasonably generic framework for studying BSM models, hence the normal time-like QCD and QED showering has been extended by the addition of the HV sector. HV being a light hidden sector, the associated particles may have masses as low as 10 GeV and the spectrum of the valley particles and their dynamics depends on the valley gauge group $G_v$, their spin and the number of particles contained in the theory, along with their group representations. There are 12 particles which are charged under both the SM and HV symmetry groups, with each particle coupling flavour-diagonally with the corresponding state in SM, but has a fundamental representation in the HV colour symmetry as well. The HV particles with no SM couplings are invisible and their presence can only be detected by observing the amount of missing transverse momentum present in a particular event. In case of the SU(N) symmetries, the gauge group remains unbroken leading to massless gauge bosons $g_v$ and there is confinement of partons. In this scenario, the HV quarks $q_v'$s and $\bar{q}_v'$s can be

obtained which can either decay back to SM or remain stable, depending upon the mixing of the states. If it is off-diagonal, flavour-charged, then the $q'_v$s can exist as stable and invisible states, whereas diagonal ones can decay back to the SM and contribute to formation of visible hadrons, leading to the formation of semi-visible jets.

Studies discussed in [7, 10–12] have shown that the decay of dark hadrons also depends on the mediator to the visible sector. Two different dark quark flavours combine to form dark $\pi^+$, $\pi^-$, $\pi^0$, and dark $\rho^+$, $\rho^-$, $\rho^0$, where the dark $\rho$'s are assumed to be produced thrice as much as pions. The dark $\rho_d$ mesons tend to decay promptly via the decay channel $\rho_d \to \pi_d \pi_d$, except for the $\rho_d^0$ meson, which decays into SM particles due to portal interactions of the mediator coupling the SM sector to the dark sector. Hence, for the jet-substructure studies, the $\rho^+$ and $\rho^-$ mesons can be treated as intermediate dark states, which subsequently decay to the $\pi_d^+$, $\pi_d^-$, $\pi_d^0$ mesons, constituting the final dark states, and the $\rho_d^0$ meson contributes to the visible fraction of the semi-visible jet.

In order to understand better the signal cross-section sensitivity in terms of the coupling strength $\lambda$, a cross-section scan was performed with $\lambda$ values in the range of [0.1,1] as shown in Fig. 3. It was observed that for $\lambda = 1$, the current model leads to a scenario where kinematic selections like that on missing transverse momentum or leading jet $p_T$ is far more effective in suppressing the background compared to the substructure observables considered here. If subsequently it is found that there are specific jet-substructure observables which can be useful in discriminating signal over background then $\lambda$ ranges of 0.6 and lower values, draws a scenario where kinematic selections still leaves roughly similar signal and multijet background contribution, and the potential discriminating power of those observables will be more important. However, in an experimental setup, the multijet contribution increases by several folds due to the presence of mis-measured jets, and hence $\lambda = 0.6$ can be treated as a average bound value, moving to relatively higher $\lambda$ values within the originally mentioned lambda range, for performing detector-level studies using jet-substructure variables.

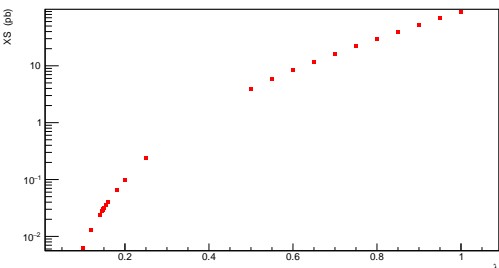

Figure 3: A cross-section scan of the semi-visible jets signal having a mediator mass of 1500 GeV, and a dark-matter candidate mass of 10 GeV with respect to $\lambda$.

# 3 Comparison of JSS observables

## 3.1 Analysis strategy

The signal samples, at $\sqrt{s} = 13$ TeV are generated by using a t-channel simplified dark-matter model in Madgraph5 [13] matrix element (ME) generator, with $xqcut = 100$ [1] and NNPDF2.3 LO PDF set [14], a mediator mass of 1500 GeV, and a dark-matter candidate mass of 10 GeV. Different $R_{inv}$ fractions result in somewhat different kinematics, so $R_{inv}$ values of 0.25, 0.50

---

[1]defined as the minimum $k_T$ separation between partons

and 0.75 are studied, as well as the values of 0 (no dark component) and 1 (fully dark jet) corresponding to the boundary conditions. The process $pp \rightarrow \chi\bar{\chi}$ with up to two extra jets is simulated and MLM matched [15] in order to have a reasonable cross-section and obtain a proper signal which does not get swamped under multijet background. Hence, the pair/associated production modes involving intermediate $\Phi$'s are generated and decayed within Madgraph5, and the MLM matching ensures reasonable relative contributions from the two classes of event shown in Fig. 1. The multijet production described by QCD are generated with Pythia8. As we are mostly concerned with the substructure of the jets, the lack of higher order matrix element in background simulation is not a concern. Multijet events were also generated with Madgraph5 generator, with up to 4 partons in ME level, showered with Pythia8 as before, but using CKKW-L [16] matching. No dramatic difference is observed in substructure observables. Pythia8 model does take into account the effect of heavy flavour jets created by gluon splitting.

It must be noted though, that while multijet events at particle level mostly have low values of missing transverse momentum, at detector level, due to mis-measurement of jet energy and direction, a large fraction of events acquire large values of missing transverse momentum. This is essentially an irreducible background. Another possible origin missing transverse momentum close to jets arise the jet area includes dead calorimeter cells. In experimental searches, this is typically accounted for by removing jets when the angular separation of jets and missing transverse momentum direction is less than $\Delta\phi < 0.4$. For a signature like this, that requirement is clearly unusable, but events where jet areas include dead calorimeter cells can be discarded, and in ATLAS this is seen to be a small fraction of events [17].

In this study, we are using large-radius jets, more specifically anti-$k_t$ [18] jets with R=1.0, trimmed (with $R_{sub} = 0.2$ and $f_{cut} = 0.05$) [19] in order to stay close to potential experimental analysis. The validity of this can be seen in Fig. 4, where it is evident that large radius jets better contain the semi-visible jets. We note that reclustered jets [20] may also be a good way to probe these events, but we leave that for a future study.

The large-radius jets are required to have $p_T > 250$ GeV. As stated in Sec. 2, the identifying signature of semi-visible jets is the alignment of the event missing transverse momentum along the direction of such a jet. Therefore we require the presence of at least one large-radius jet within $\Delta\phi < 1.0$ of the missing transverse momentum direction, and that jet is tagged as a svj. Additionally, we require at least 200 GeV of missing transverse momentum, owing to the fact that an actual search using a missing transverse energy trigger will require that threshold.

It is however interesting to note that in a majority of events, the subleading jet in transverse momentum is tagged as the svj, as can be see from the distribution of $\Delta\phi$ between leading and subleading jets with the missing transverse momentum direction in Fig. 5.

In Fig. 6, we show that the events with svj have high missing transverse momentum compared to the background jets, as expected, and also the $p_T$ distribution of the svj with the background jets. We pick the leading large-radius jet, without any requirement on missing transverse momentum as the background jet. Here we note that even though the svj is more often than not the sub-leading jet, we are mostly interested in differentiating svj from standard quark/gluon-initiated jets, so we can use the leading jet from the background without any loss of generality. It was observed that using only quark or only gluon initiated background jets made no difference.

Apart from the multijet process, which is the dominant background, $W/Z$ + jets processes can contribute to the background. However, the processes with one or more leptons can be rejected by vetoing events with leptons. The $W \rightarrow \tau_{had}\nu$ and $Z \rightarrow \nu\nu$ processes result in irreducible backgrounds, but since the large radius jets will still come from quarks or gluons, the considerations for multijet events still hold. A detailed study of these backgrounds is left for future work.

## 3.2 JSS observables

Many jet substructure observables have been designed over last decade or so [21, 22], with different sensitivity to different signal jets. In recent works, the focus was on energy correlation observables [23], and discussed the non trivial theoretical uncertainties associated with jet substructure. In this study, we looked at a broad array of observables, Les Houches angularity (LHA) [24], splitting variables $r_g$ and $z_g$ [25, 26], N-subjettiness ratios, $\tau_{21}$ and $\tau_{32}$ [27], and the ratios of energy correlation functions, $C_2$, $D_2$, ECF2, and ECF3.

LHA is the case where the exponents $\kappa = 1, \beta^{\mathrm{LHA}} = 0.5$ in the generalised angularity expression:

$$\lambda^{\kappa}_{\beta^{\mathrm{LHA}}} = \sum_{i \in J} z_i^{\kappa} \theta_i^{\beta^{\mathrm{LHA}}},$$

where $z_i$ is the transverse momentum of jet constituent $i$ as a fraction of the scalar sum of the $p_{\mathrm{T}}$ of all constituents and $\theta_i$ is the angle of the $i^{\mathrm{th}}$ constituent relative to the jet axis, normalised by the jet radius.

Energy correlation functions ECF2 and ECF3 [28], and related ratios $C_2$, $D_2$ [29] for a jet $J$ are derived from:

$$\mathrm{ECF1} = \sum_{i \in J} p_{\mathrm{T}_i},$$

$$\mathrm{ECF2}(\beta^{\mathrm{ECF}}) = \sum_{i < j \in J} p_{\mathrm{T}_i} p_{\mathrm{T}_j} \left( \Delta R_{ij} \right)^{\beta^{\mathrm{ECF}}},$$

$$\mathrm{ECF3}(\beta^{\mathrm{ECF}}) = \sum_{i < j < k \in J} p_{\mathrm{T}_i} p_{\mathrm{T}_j} p_{\mathrm{T}_k} \left( \Delta R_{ij} \Delta R_{ik} \Delta R_{jk} \right)^{\beta^{\mathrm{ECF}}},$$

where the parameter $\beta^{\mathrm{ECF}}$ weights the angular separation of the jet constituents. In this analysis, $\beta^{\mathrm{ECF}} = 1$ is used, and for brevity, $\beta^{\mathrm{ECF}}$ is not explicitly mentioned hereafter. The ratios of some of these quantities (written in an abbreviated form) are defined as :

$$e_2 = \frac{\mathrm{ECF2}}{(\mathrm{ECF1})^2},$$

$$e_3 = \frac{\mathrm{ECF3}}{(\mathrm{ECF1})^3}.$$

These ratios are then used to generate the variable $C_2$ [28], and its modified version $D_2$ [29, 30], which have been shown to be particularly useful in identifying two-body structures within jets [31]:

$$C_2 = \frac{e_3}{(e_2)^2},$$

$$D_2 = \frac{e_3}{(e_2)^3}.$$

The $N$-subjettiness describes to what degree the substructure of a given jet is compatible with being composed of $N$ or fewer subjets. In order to calculate $\tau_{\mathrm{N}}$, first $N$ subjet axes are defined within the jet by using the exclusive $k_{\mathrm{t}}$ algorithm, where the jet reconstruction

continues until a desired number of jets are found. A parameter $\beta^{\text{NS}}$ gives a weight to the angular separation of the jet constituents. In the studies presented here, the value of $\beta^{\text{NS}} = 1$ is used.

Among these observables, we have primarily focused on $C_2$, LHA and $\tau_{21}$, and $\tau_{32}$ for this study. In general we noticed that $D_2$ and ECF2 were fairly similar, but were less sensitive as compared to $C_2$, and ECF3, $r_g$ and $z_g$ were mostly insensitive to the effect we are probing.

In order to compare signal and background large-radius jets with similar kinematics, we look at two different jet $p_{\text{T}}$ ranges, 400–600 GeV and 800–1000 GeV, motivated by Fig. 6.

## 3.3  Results

Distributions of several jet substructure observables are compared between semi-visible and ordinary jets in Fig. 7. The results in $p_{\text{T}}$ range of 400–600 GeV are shown, but the results in the 800–1000 GeV range exhibit the same feature, albeit with a lack of statistics. The distributions are normalised to area, not to cross-section, as we are interested in probing the shape differences.

The overall interpretation is, semi-visible jets result in more multi-pronged substructure, as evidenced in higher values of $C_2$ and LHA. For $\tau_{21}$, and $\tau_{32}$, the lower values of signal indicate that those are more multi-pronged respectively, whereas the background is more single pronged. LHA, surprisingly does not show any difference when changing $R_{\text{inv}}$. For $\tau$, lower $R_{\text{inv}}$ values seem closer to background, indicative of the the fact that lower dark hadron fraction is less multi-prong, and indeed more background like. It is important here to note that the N-subjettiness and ECF variables have somewhat different design philosophy [32]. While the former strongly depend on the axis, and are more sensitive to determine if the jet has at least N-prongs, the latter are more sensitive to separating one or two prong substructures. Therefore, if the svj is more multi-pronged than two-pronged, then these two classes of observables can appear to show contradictory characteristics, i.e. $\tau_{21}$ will indicate that the svj is atleast two pronged, whereas $C_2$ will state it is not two-pronged. For example, $C_2$ for the signal with highest $R_{\text{inv}}$ fraction appear closest to the background, in apparent contradiction with $\tau_{21}$, which just implies that it is rather different from being two-pronged, not necessarily single-pronged.

The results here are shown without any theoretical systematic uncertainty. Based on the recent study [23], we can very conservatively assume a 30–40% flat uncertainty on these substructure variables. That would not make the general conclusions arrived at this article invalid, but for certain observables, like $\tau_{21}$ for lower $R_{\text{inv}}$ values, the discrimination power would be degraded. Also, detector effects can degrade the performance as well, but a quick check using parametrised smearing [33] showed the results we obtain are robust. The effect of pile-up was not considered as well, but use of groomed jets (trimming was used here) should mitigate the effect of it to a large extent. Smearing of the substructure variables makes the peaks somewhat diffused and the difference between the signal and background slightly less pronounced.

# 4  Understanding the differences

## 4.1  Model dependence

Currently the only dark shower model that can be used to simulate semi-visible jets is the Pythia8 Hidden Valley module, as discussed in Sec. 2. So an obvious concern is, to what extent the differences seen between signal and background in the previous section is model-dependent. Due to the absence of another model, an unambiguous answer to this question

is difficult to arrive at, but considering an extreme scenario of $R_{\text{inv}} = 0$ might offer us some clues. Imposing this condition implies that our signal large-radius jets consist entirely of visible hadrons, and subsequently the behaviour is expected to be like background jets, with low missing transverse momenta, as seen in Fig. 6.

However, in this case, requirements on missing transverse momentum magnitude and direction does not really make sense for signal, so for these comparisons, a background-like event selection is employed, assuming leading large-radius jet is the svj.

Considering a background-like event selection, along with the $R_{\text{inv}} = 0$ condition, if the substructure of the signal jets resemble that of the background jets, then that would give us some confidence that the difference seen for non-zero $R_{\text{inv}}$ values, as seen before, are caused not only by the model specifications but also involve the effects owing to the dark hadrons. The two parameters controlling the HV shower, that were expected to to be most consequential for this study, are defined in Table 1.

Table 1: Hidden Valley model parameters considered in the study. The HV fixed alpha scale corresponds to $\alpha_{strong}$ of QCD or $\alpha_{em}$ of QED.

| Observable | Pythia8 name | Indicated in text by: |
|---|---|---|
| Fixed alpha scale of gv/gammav emission | *HiddenValley:alphaFSR* | $\alpha_{\text{HV}}$ |
| Lowest allowed $p_{\text{T}}$ of emission | *HiddenValley:pTMin* | $pTmin_{\text{HV}}$ |

We have found minimal dependence on $pTmin_{\text{HV}}$ (which was also fairly independent of dark-hadron mass scale), but in Fig. 8, we see how the substructure variables change significant with the variation of $\alpha_{\text{HV}}$, where other intermediate values were also probed, but are not shown.

The takeaway message is that in signal jets, $C_2$ can be made to look similar to background jets for $\alpha_{\text{HV}} = 0.1$. The trend for LHA is not so clear, and the tau observables have the weakest dependence, indicating the latter is not affected by the HV model implementations, so they will be looked into more carefully as we go along.

While this $\alpha_{\text{HV}}$ value is the closest to the QCD $\alpha_{FSR}$ value used in generators, one must note that they cannot be treated at the same footing, as QCD coupling is run at 2-loops. However, based on these results, we will use this $\alpha_{\text{HV}}$ value in the rest of the comparisons.

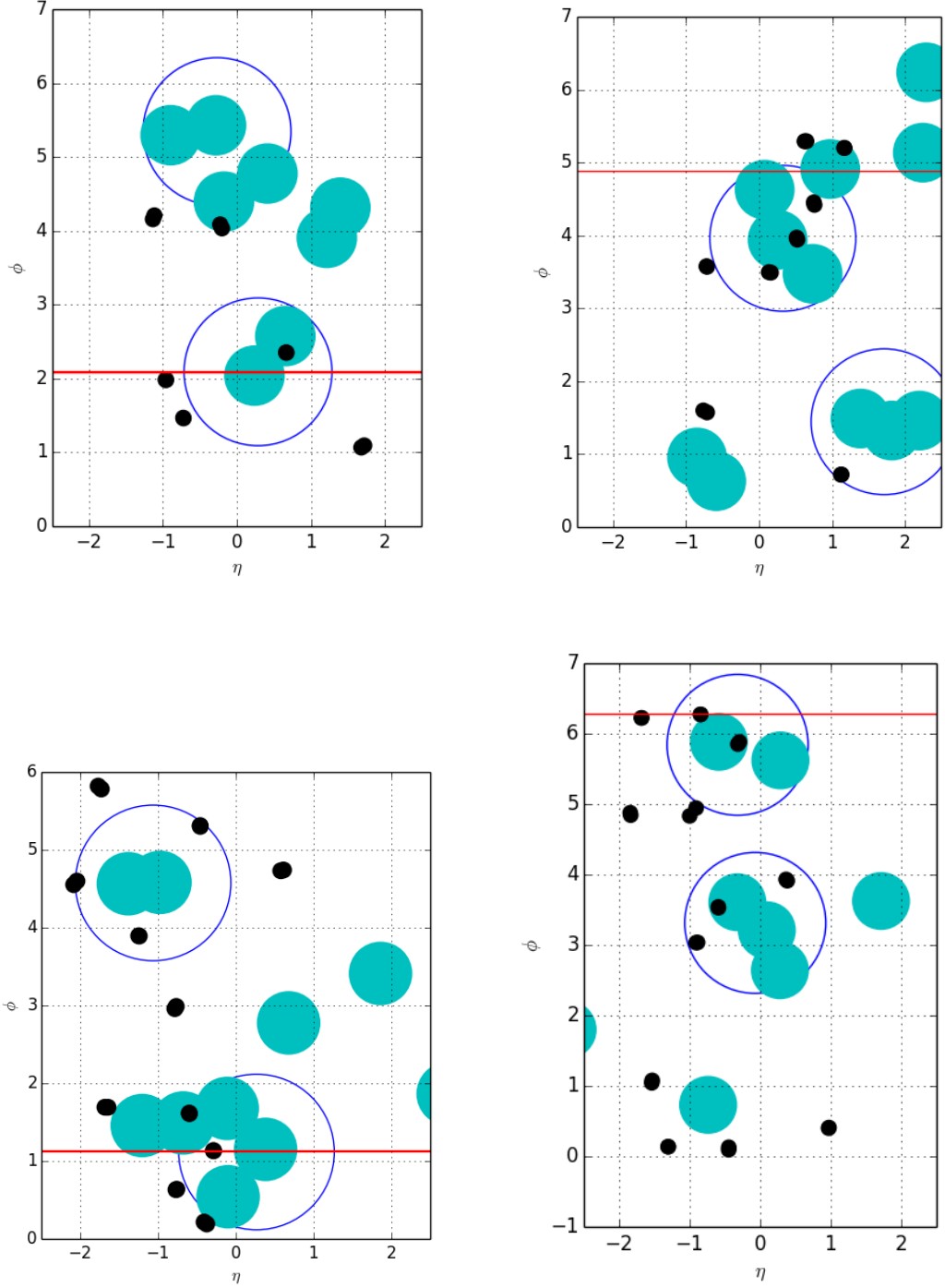

Figure 4: Various objects are plotted in $\eta - \phi$ plane for four representative events. The large hollow blue circles represent the trimmed large-radius jets, the filled cyan circles represent anti-$k_{\mathrm{t}}$ jets with R=0.4, the black points represent dark hadrons, and the red line the direction of missing transverses momentum.

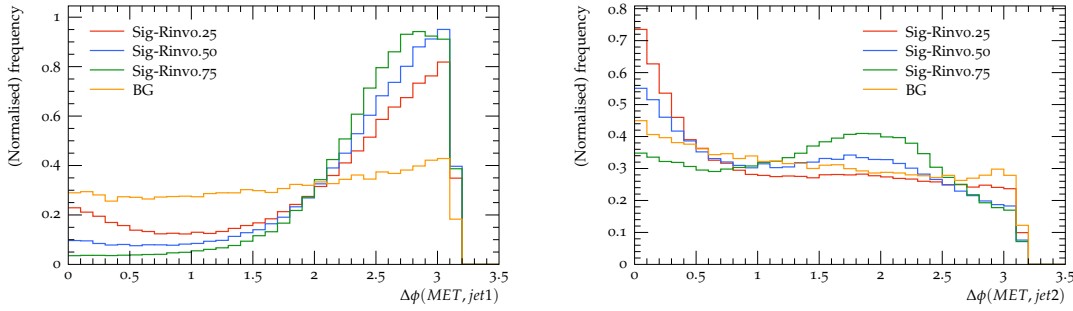

Figure 5: Distributions of the azimuthal angle difference between the leading (left) and subleading (right) jets with the direction of missing transverse momentum for three different signals corresponding to $R_{\rm inv}$ values of 0.25, 0.50 and 0.75, and the background.

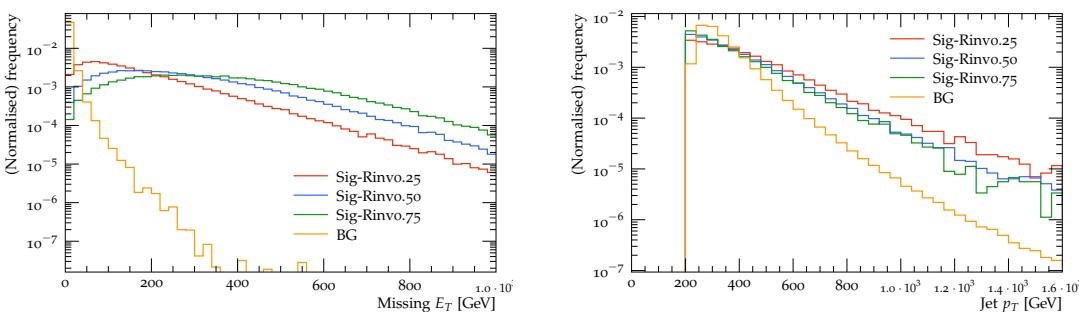

Figure 6: Distributions of MET (right) and leading jet $p_{\rm T}$ (left) for three different signals corresponding to $R_{\rm inv}$ values of 0.25, 0.50 and 0.75, and the background.

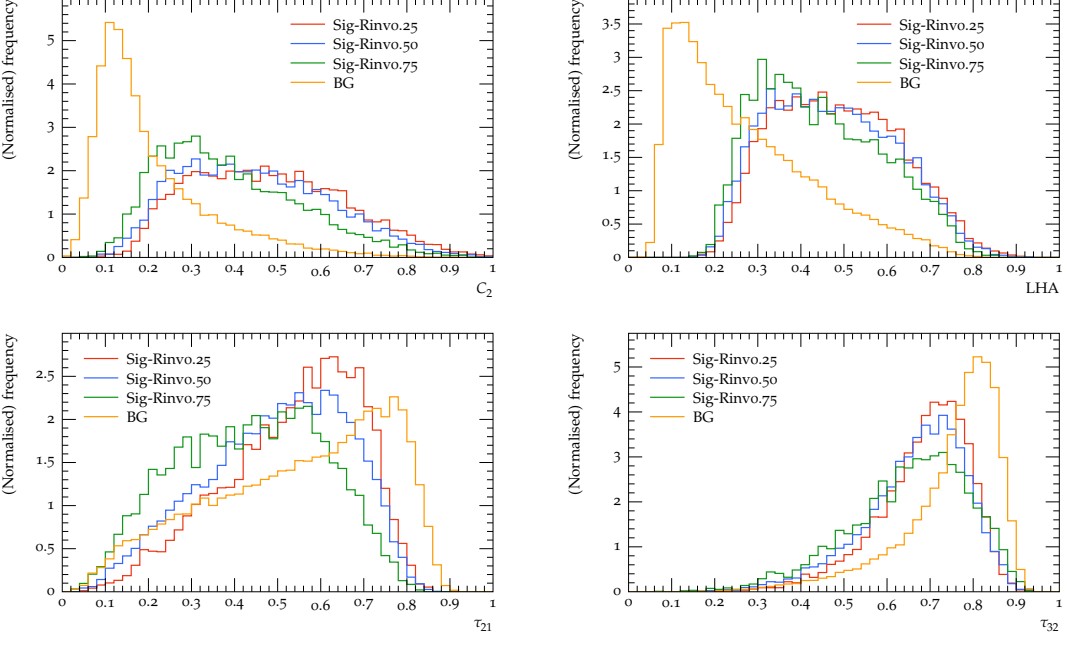

Figure 7: Comparisons of $C_2$ (top left), LHA (top right), $\tau_{21}$ (bottom left) and $\tau_{32}$ (bottom right) between three different signals corresponding to $R_{\rm inv}$ values of 0.25, 0.50 and 0.75, and the background for $\alpha_{FSR} = 1$ and $p_T = 400 - 600$ GeV

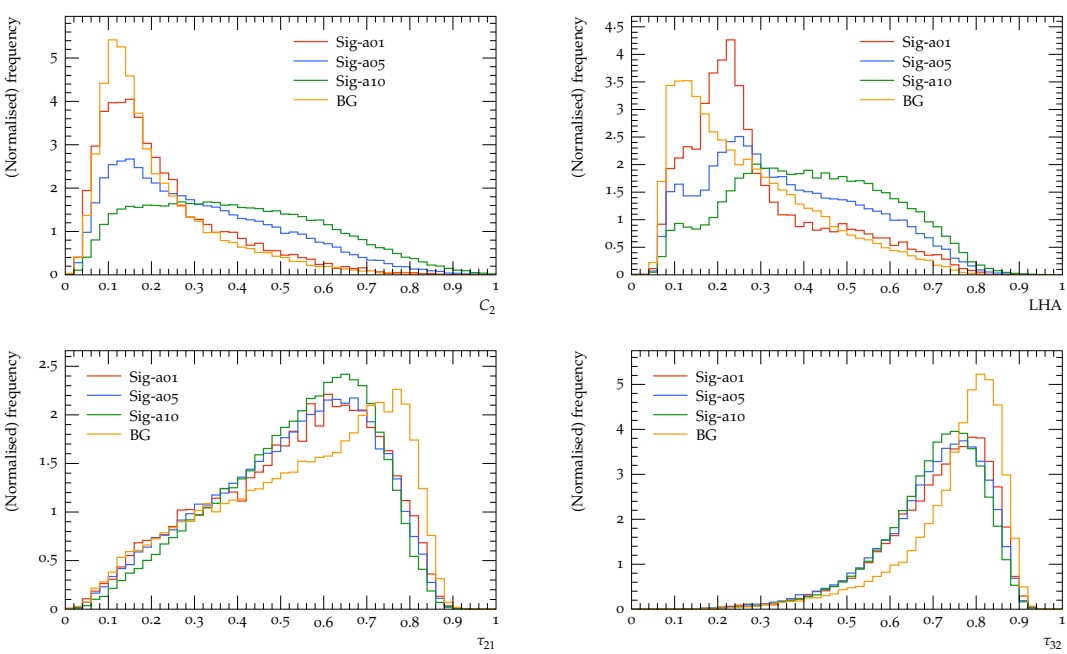

Figure 8: Comparisons of $C_2$ (top left), LHA (top right), $\tau_{21}$ (bottom left) and $\tau_{32}$ (bottom right) between three different signals corresponding to $\alpha_{\mathrm{HV}}$ values of 0.1, 0.5 and 1.0 with $R_{\mathrm{inv}} = 0$, $p_{\mathrm{T}}$ range 400–600 GeV, and the background. It is interesting to note that that a signal with $R_{\mathrm{inv}} = 0$ is not necessarily equivalent to the background.

## 4.2 Origin of the differences

An understanding of the observed behaviour of jet substructure observables in semi-visible jets is last piece of the puzzle. In order to investigate this, we asked three questions:

1. What is effect of initial state radiation (ISR) and extra radiation on jet substructure?

2. Does decay from intermediate to final dark hadrons affect the substructure?

3. How does grooming affect jet substructure in svj?

In order to answer these, we turn to the other extreme scenario of $R_{\text{inv}}= 1$, which corresponds to the case where the signal jet consists entirely of dark hadrons. Evidently in this case the signal jet itself is ill-defined, but by considering the unphysical scenarios of using dark hadrons in jet clustering, we can try to disentangle several effects.

First, the dark hadrons can be used to form signal jets, along with visible hadrons or without visible hadrons. The extra ME jets and the ISR can be turned off in either case. In each case, the leading large-radius jet is taken, and unless otherwise mentioned, comparisons are performed in the $p_{\text{T}}$ range of 400–600 GeV. We look at the same observables as before in Fig. 9.

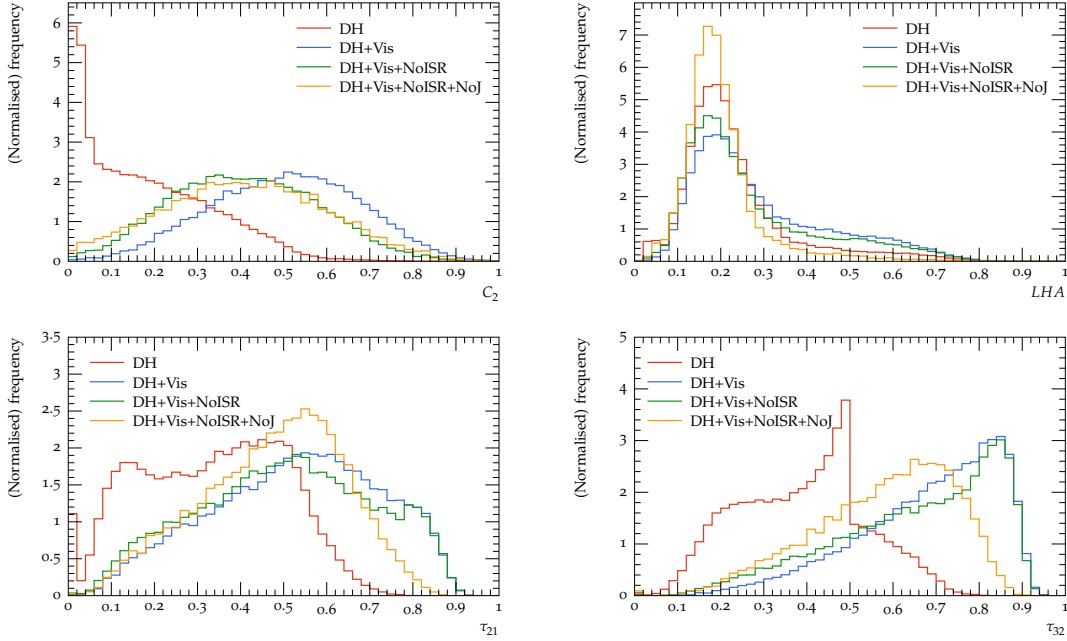

Figure 9: Comparisons of $C_2$ (top left), LHA (top right), $\tau_{21}$ (bottom left) and $\tau_{32}$ (bottom right) between different signals corresponding to clustering only with dark hadrons (DH), adding visible hadrons (Vis), tuning ISR off (NoISR), and also turning extra ME jets off (NoJ).

Clustering only dark hadron in jets is indicative of the shape an ideal semi-visible jet may result in. The more realistic scenario is of course clustering the visible hadrons. In $R_{\text{inv}} = 1$ scenario considered here, the visible hadrons come almost exclusively from ME level extra jets and ISR. Looking at $C_2$ and $\tau$ observables, its clear that adding visible hadrons make the signal jets less multiprong, by filling in the gaps. As before higher $C_2$ values indicate moving away from two-prong structure.

It is interesting to see how the visible hadrons coming from ISR and ME extra jets affect the substructure differently. Turning off the ISR affects $C_2$ more than $\tau$ observables, perhaps

indicating the $C_2$ is more sensitive to the softer radiation. Additionally turning ME extra jets off has the opposite behaviour, it does not affect $C_2$, but makes taus indicative of slightly more two/three pronged substructure. It also implies that ISR adds more activity to semi-visible jets compared to ME extra jets, making them slightly more multi-pronged. This may be due to the fact ISR jets are more isotropic so they can overlap with SVJ, while ME jets are more well separated. Turning off ME extra jets makes the svj produced with less $p_T$, so that implies we are not comparing the *same* jets in these cases. Surprisingly LHA seem rather insensitive.

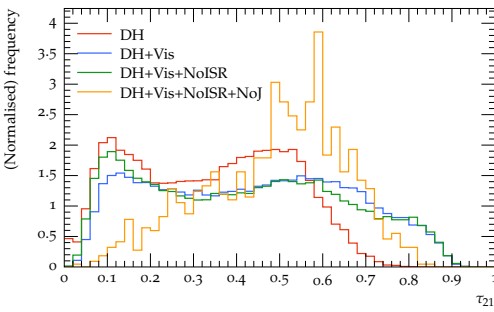
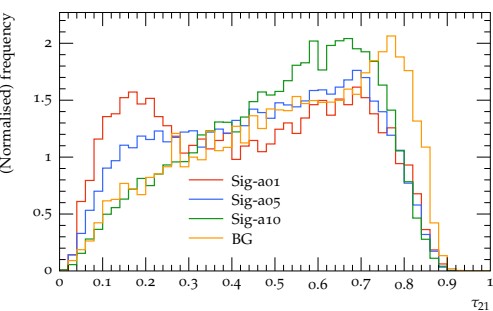

Figure 10: Comparisons of $\tau_{21}$ in jet $p_T$ 800–1000 GeV range between different signals corresponding to clustering only with dark hadrons, adding visible hadrons, tuning ISR off, and also turning extra ME jets off with $R_{inv} = 1$ (left) and for three different signals corresponding to $\alpha_{HV}$ values of 0.1, 0.5 and 1.0 with $R_{inv} = 0$ (right).

An interesting feature can be seen the bottom left $\tau_{21}$ distribution of Fig. 9, where two peaks appear. This feature in enhanced for the higher $p_T$ range, and also appears for lower values of $\alpha_{HV}$ as discussed in Sec. 4.1, which can be seen in Fig. 10. This is independent of adding SM hadrons, except when ME extra jets are turned off. This observation is consistent with the occurrence of this feature with higher $p_T$, where jets can be more collimated and two-pronged. The lower values of $\alpha_{HV}$ similarly indicate less radiation.

Another sanity check is to examine if the decay from intermediate dark hadrons to the final dark hadrons considered above is responsible for creating or enhancing the substructure. We make the intermediate dark hadrons stable, and cluster them in jets, with and without visible hadrons. In Fig. 11, the comparison of those with the previous results show essentially no difference, except a slightly more flattish shape in lower values of tau for the current case. So it is safe to say the observed substructure is not due to HV decay structure.

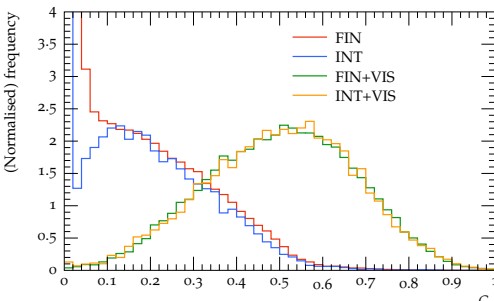
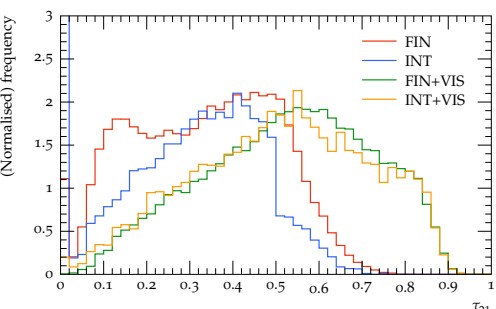

Figure 11: Comparisons of $C_2$ (left) and $\tau_{21}$ (right) between different signals corresponding to clustering only with final dark hadrons (FIN), intermediate dark hadrons (INT) and adding visible hadrons (VIS) in both cases. The entries at zero correspond to cases where the substructure variable could not be calculated, as only in rare cases the actual value of the observable was zero.

The next test was how grooming affects the substructure of semi-visible jets, as grooming preferentially cuts out soft or wide angle radiation. We test the effect of trimming here.

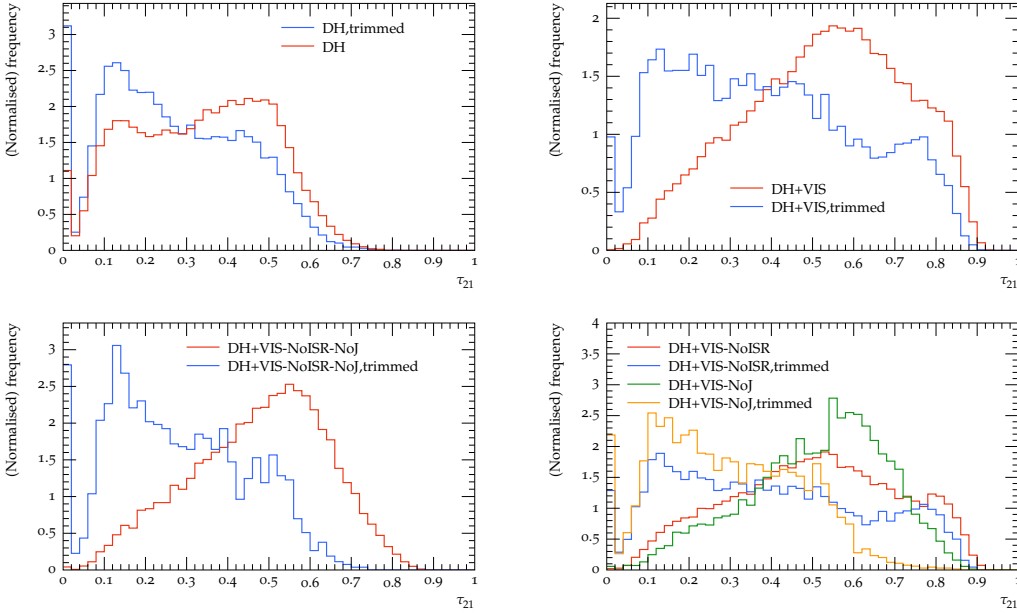

Figure 12: Comparisons of $\tau_{21}$ (right) between ungroomed and trimmed case, for signal configurations corresponding to clustering only with dark hadrons (top left), dark and visible hadrons (top right) turning extra ME jets and ISR off (bottom left) and turning one (but not both) of them off (bottom right). The entries at zero correspond to cases where the substructure variable could not be calculated.

In Fig. 12, we compare different configurations with and without trimming. Trimming in general moves $\tau_{21}$ to the left, indicating a cleaner two pronged substructure. This is least pronounced for 'only dark hadron' case, slightly more when visible hadrons are also clustered, and most pronounced for no extra ME jets or ISR case. A comparison between the scenarios of no extra ME jet and no ISR indicates ISR gets more affected by trimming. The same conclusion could also have been reached at looking at $C_2$, but the effect was less pronounced. Trimming did not affect the $p_T$ spectra of the signal jets.

Last but not the least, after exploring what effects are not responsible for the specific substructure of semivisible jets, we are ready to answer what is actually responsible. For finite $R_{inv}$ values, only the visible hadrons are clustered in jets in Sec. 3.3, and slightly different substructure were seen for different $R_{inv}$ values. Now, as in Sec. 4.2, if the final dark hadrons are also clustered in the jets, we should expect this difference to go away, as the different amount of *missing* hadrons in each case presumably was responsible for the difference. Indeed, in Fig. 13, we see the expected behaviour. For $C_2$, the lines corresponding to the cases where dark hadrons are clustered are almost identical, and while they are not identical for $\tau_{21}$, they lie in between the two original lines. This indicate that the substructure becomes less two-pronged with visible and dark hadrons in them, and the absence of the dark hadrons create the two-pronged structure. These distributions were made with $\alpha_{HV} = 1$ to have the maximum possible dark radiation.



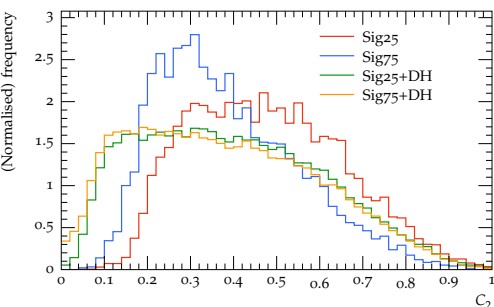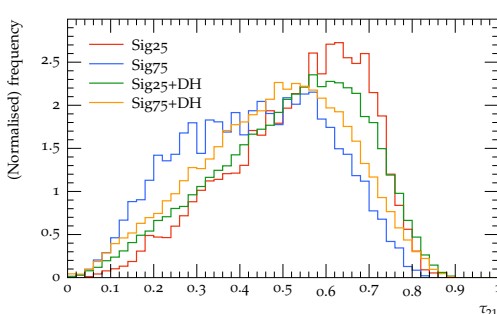

Figure 13: Comparisons of $C_2$ (left) and $\tau_{21}$ (right) for different signals corresponding to $R_{\text{inv}}$ values of 0.25 and 0.75, clustering only the visible hadrons and clustering also with final dark hadrons.

# 5 Conclusions

A comprehensive study of the substructure of semi-visible jets has been performed. We demonstrated that specific hidden valley parameter configurations can reduce the dark shower model dependent features of the signal jets. The origin of the substructure in semi-visible jet is neither caused by the decay of intermediate dark hadrons, nor by extra ME jets, or ISR, although the latter two affect the substructure. The substructure is created by the interspersing of visible hadrons with dark hadrons. The substructure observables which are least affected by model dependence can be used in searches, and also as inputs to machine learning algorithms trying to identify semi-visible jet via anomaly detection [34–41], assuming the relatively similar contribution from signal and background processes.

# Acknowledgements

We thank Tim Cohen, Siddharth Mishra-Sharma and Joel Doss for many useful ideas and suggestions. We also thank Caterina Doglioni, Jannik Geisen, Eva Hansen for many interesting discussions, and for organising the Lund darkjets mini workshop in November 2019, where the authors benefited from discussions with Rikkert Frederix, Stefan Prestel and Torbjörn Sjöstrand. We thank Andy Buckley and Marvin Flores for going over the draft and making relevant suggestions.

**Funding information**  DK is funded by National Research Foundation (NRF), South Africa through Competitive Programme for Rated Researchers (CPRR), Grant No: 118515. SS thanks University of Witwatersrand Research Council for Sellschop grant and NRF for grant-holder linked bursary.

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
