# Peer review of "Exploring Jet Substructure in Semi-visible jets"

_SciPost Physics, doi:SciPost Phys. 10, 084 (2021)_

## Round 3 · Referee Report · Jonathan Butterworth (Referee 1) · 2021-1-3

Strengths

The paper uses state-of-the-art jet substructure variables to confront the experimentally challenging and phenomenologically interesting topic of semi-visible jets.

The method is clearly documented and the code used is open source and widely available so the results should be readily reproducible.

The main conclusion, that interesting jet substructure is present in these jets and that is primarily caused by the invisible shower, not by either dark hadron decay or additional QCD radiation, seems substantiated and to be of quite general interest.

Weaknesses

There are places where important conclusions of sub-studies the authors have carried out are stated without giving evidence or enough detail.

The study relies on a single implementation of a dark shower model, not carried out by the authors. (However efforts are made in the study to address this weakness.)

There are number of slips in the language, which may be typos but which in places do interfere with the understanding.

Report

Since the authors have brought experimental expertise in jet substructure to bear on a new area, I think the paper could arguably be said to meet two of the criteria for the journal:

  1. Open a new pathway in an existing or a new research direction, with clear potential for multipronged follow-up work;

  2. Provide a novel and synergetic link between different research areas.

However, I think the strength of this "opening" and "linking" depends on the responses to the comments, and as it stands, it is not publishable in SciPost physics. With sufficiently compelling/complete responses, this opinion would likely change.

Requested changes

  1. On the face of it, heavy flavour jets (especially charm), which more often contain neutrinos, would be a background of concern, but they are not discussed, or studied (unless g --> ccbar included in the shower? if so maybe this would be the dominant source anyway). The authors should either study them, or else explain their rationale for neglecting them, and estimate the impact.

  2. Is the background only 2->2 QCD, or are 4 jet MLM matched configurations included? Please clarify/comment on the impact.

  3. On p5 the authors state that it makes no difference whether all quark or all gluon initiated jets are used. This is surprising. Can you give more detail on what evidence led to this conclusion?

  4. The paragraph at the top of p6 is an example of confusing grammar, especially the sentence with "actually not the case". What is really meant here? Can you write it more clearly?

  5. Please provide more detail/evidence on the "quick check" of detector effects.

  6. Related to 5, was pile up considered? Can you comment on its potential impact?

  7. I think I understood the paragraph at the top of page 8, but only after reading the later section. It is very unclear, can you please try to clarify it? This would seem to be quite critical to the overall conclusion.

  8. The second sentence of the conclusions seems to say that by choosing specific model parameters you can reduce the model dependence. I think I can work out what is intended, but as written it doesn't really make sense. Please clarify.

  • validity: good
  • significance: high
  • originality: high
  • clarity: good
  • formatting: excellent
  • grammar: reasonable

Author:  Deepak Kar  on 2021-01-22  [id 1171]

(in reply to Report 1 by Jonathan Butterworth on 2021-01-03)

We thank the referee for this thorough review of the paper (specially during the holidays), and suggesting ways how the physics message and the presentation can be improved. We are glad that they feel that the paper explores a new area, which is exactly what authors intended, even though we completely agree that this is a first look at this, which should serve as a baseline for further work.

Please find our responses inline.

Requested changes

  1. On the face of it, heavy flavour jets (especially charm), which more often contain neutrinos, would be a background of concern, but they are not discussed, or studied (unless g --> ccbar included in the shower? if so maybe this would be the dominant source anyway). The authors should either study them, or else explain their rationale for neglecting them, and estimate the impact.

That is an interesting point. We used “standard” Pythia shower: HardQCD:all = on, which includes HardQCD:gg2ccbar as mentioned in Pythia manual: http://home.thep.lu.se/~torbjorn/pythia82html/QCDProcesses.html

We definitely agree that a LL generator like Pythia would probably not model the kinematics correctly. In order to convince ourselves, we generated just gg->ccbar events, and the MET distribution is attached is slightly steeper, but JSS observables are almost identical.

So we think it is reasonable to say while semileptonic decays of charm can be a dominant background, with a MET requirement of 200 GeV, we can reduce it significantly. We added: Pythia8 simulation does take into account the effect of heavy flavour jets created by gluon splitting.

  1. Is the background only 2->2 QCD, or are 4 jet MLM matched configurations included? Please clarify/comment on the impact.

It is 2->2 QCD by pure Pythia8. In order to address this concern, we generated background events with MG5+Pythia8, MLM matched, which is how the signal was generated. A comparison with current background distributions showed no significant difference, which we think is expected as the we are using a fairly large jet pT bin, and the shower is mostly responsible for generating the substructure.

We added: As we are mostly concerned with the substructure of the jets, the lack of higher order matrix element in background simulation is not a concern.

  1. On p5 the authors state that it makes no difference whether all quark or all gluon initiated jets are used. This is surprising. Can you give more detail on what evidence led to this conclusion?

Indeed, we were surprised as well, and in fact this is one of the first things we checked. We are afraid we used Pythia as well, using HardQCD:gg2gg HardQCD:qqbar2gg And HardQCD:gg2qqbar HardQCD:qq2qq In each case. We are however aware that gluon jet modelling is less constrained in most generators including Pythia [1].

[1] https://arxiv.org/abs/1704.03878

  1. The paragraph at the top of p6 is an example of confusing grammar, especially the sentence with "actually not the case". What is really meant here? Can you write it more clearly?

Rephrased as: “Therefore, if the svj is more multi-pronged than two-pronged, then these two classes of observables can appear to show contradictory characteristics, i.e. tau21 will indicate that the svj is at least two pronged, whereas ECF2 will state it is not two-pronged.”

  1. Please provide more detail/evidence on the "quick check" of detector effects.

Absolutely. Since the study used Rivet, it was straightforward for us to implement the detector smearing proposed in [2]. While that made the differences between signal and background slightly less pronounced, the main conclusions were not affected. We added a line in text: Smearing of the substructure variables makes the peaks somewhat diffused and the difference between the signal and background slightly less pronounced. [2] https://arxiv.org/abs/1910.01637

  1. Related to 5, was pile up considered? Can you comment on its potential impact?

We are afraid not, however we did use trimmed large-radius jets, which should mitigate the effect of soft-radiation to a certain extent. It is a bit challenging to simulate pileup at particle level, but a priori we would think it will have the same effect of detector smearing, i.e “diffuse” the distributions somewhat, but would not dramatically shift the peaks.

  1. I think I understood the paragraph at the top of page 8, but only after reading the later section. It is very unclear, can you please try to clarify it? This would seem to be quite critical to the overall conclusion.

Rephrased as: “Considering a background-like event selection, along with the \rinv$=0$ condition, if the substructure of the signal jets resemble that of the background jets, then that would give us some confidence that the difference seen for non-zero \rinv values, as seen before, are caused not only by the model specifications but also involve the effects owing to the dark hadrons.”

  1. The second sentence of the conclusions seems to say that by choosing specific model parameters you can reduce the model dependence. I think I can work out what is intended, but as written it doesn't really make sense. Please clarify.

Rephrased as: “ We demonstrated that specific hidden valley parameter configurations can reduce the dark shower model dependent features of the signal jets.”

Thanks again, Sukanya and Deepak

---

## Round 3 · Referee Report · Tilman Plehn (Referee 2) · 2021-1-13

Strengths

1- the paper tackles a relevant and timely problem, namely to search for dark matter or other new physics outside the hard scattering; 2- it's actually a very hard problem; 3- the results are documented very well.

Weaknesses

1- the focus on high-level substructure variables is a little behind the wave, technically; 2- it remains unclear if there is such a thing as a best-suited observable basis and how more observables might help.

Report

The paper makes for an interesting contribution to an active and relevant field. It is not a break-through, but it is definitely worth publishing. It also defines a very nice starting point for future studies, because it is very well documented.

Requested changes

1- for the model, please give a Lagrangian or some appropriate definition in the beginning; 2- along the same line, Sec. 4.1 includes too much code-related slang for most readers to follow; 3- where is \lambda defined, before it appears in the last paragraph on p.3? 4- in the discussion of Fig.4 the authors talk about the leading/sub-leading svj, what does this mean in terms of physics? 5- on p.5 the authors start a paragraph with `Among these observables...', and I a not sure I see where this argument is coming from; 6- at the very end of Sec. 3 the authors mention detector effects. For low-level variables we have found that detector effects can make a huge difference, why is this different here. Please document as well; 7- the discussion of IRS and ME jets on p.8 is very interesting, but I have to admit that I do not understand the physics reason for the different impact, please explain that or elaborate. Could that be related to heavy-flavor jets? 8- little thing, DH in Fig.8 is not defined (as far as I can tell); 9- please cite some of the high-level or low-level analyses which were done before Ref.[31]. Not sure how many there were, but we certainly did one with our autoencoder, also referenced in Ref.[31].

  • validity: high
  • significance: good
  • originality: good
  • clarity: high
  • formatting: excellent
  • grammar: perfect

Author:  Sukanya Sinha  on 2021-01-22  [id 1172]

(in reply to Report 2 by Tilman Plehn on 2021-01-13)
Category:
answer to question
correction

We thank Tilman for this review, and suggesting ways how the physics message and the presentation can be improved. We are glad that he feel that the paper explores a new area, which is exactly what authors intended, even though we completely agree that this is a first look at this, which should serve as a baseline for further work. We must also admit that we really like the Scipost experience!

Report The paper makes for an interesting contribution to an active and relevant field. It is not a break-through, but it is definitely worth publishing. It also defines a very nice starting point for future studies, because it is very well documented. Requested changes 1- for the model, please give a Lagrangian or some appropriate definition in the beginning;

Since the model was not ours to begin with, we thought it's best to contact Tim Cohen, and he suggested that apart from the description we have listed in pg2, for further clarification we can mention that when we refer to a “SU(2)_D” gauge theory with two fermionic states, this is just a canonical Lagrangian for fermions with covariant derivatives. Does this answer the question you had, or maybe we misinterpreted your comment?

2- along the same line, Sec. 4.1 includes too much code-related slang for most readers to follow;

We have tried to clean this up with the aid of a table and shorthand names for the parameters.

3- where is \lambda defined, before it appears in the last paragraph on p.3?

It has been briefly defined in the 2nd paragraph of the introduction. Are you looking for something else?

4- in the discussion of Fig.4 the authors talk about the leading/sub-leading svj, what does this mean in terms of physics?

Ideally we should get 2 svj along with normal jets. However if the svj are exactly back to back, there is no possibility of getting real/large values of MET. But, wrt svj typically we have noticed that usually the subleading jet turns out to be the svj, as can be seen in the adjacent cartoon, and we get a significant MET contribution to balance out the momentum conservation. [Please see attached cartoon {svj-topo.png}]

5- on p.5 the authors start a paragraph with `Among these observables...', and I am not sure I see where this argument is coming from;

Rephrased as “Among these observables, we have primarily focused on ECF2, LHA and tau21, and tau32 for this study.”

6- at the very end of Sec. 3 the authors mention detector effects. For low-level variables we have found that detector effects can make a huge difference, why is this different here. Please document as well;

Absolutely. Since the study used Rivet, it was straightforward for us to implement the detector smearing proposed in [1]. While that made the differences between signal and background slightly less pronounced, the main conclusions were not affected. We added a line in text: Smearing of the substructure variables makes the peaks somewhat diffused and the difference between the signal and background slightly less pronounced. [1] https://arxiv.org/abs/1910.01637

7- the discussion of ISR and ME jets on p.8 is very interesting, but I have to admit that I do not understand the physics reason for the different impact, please explain that or elaborate. Could that be related to heavy-flavor jets?

We were also surprised by the observation. While HF jets can definitely contribute, our guess is that ISR jets are more isotropic so can overlap with SVJ, while ME jets are more well separated. We added this in the text.

8- little thing, DH in Fig.8 is not defined (as far as I can tell);

Defined in Fig8 caption now.

9- please cite some of the high-level or low-level analyses which were done before Ref.[31]. Not sure how many there were, but we certainly did one with our autoencoder, also referenced in Ref.[31].

We have added the relevant references from ref 31, in the conclusions paragraph.

Attachment:

---

## Round 3 · Referee Report · Anonymous (Referee 3) · 2021-2-19

Strengths

1) Paper explores a process in which hadronic jets are produced in proton-proton collisions but contain DM particles that cannot be detected. This is a relatively new idea and is not well investigated at the LHC

2) Paper addresses the use of jet substructure methods can be used to separate the signal (semi-visible jets = jets containing DM) from the background processes (jets containing SM particles only)

Weaknesses

1) It is unclear whether the main backgrounds are properly investigated. The signal consists of jets containing visible particles (that can be reconstructed) and invisible particles (the DM, cannot be reconstructed). The "background" is taken to be multijet production. However, Z+jets and W+jets processes can give a signal that is more similar to the signal than the multijet process. For example, - a Z-boson that recoils against a high-pt jet could produce the appropriate signal-like signature when the Z decays to tau-antitau. It would be appropriate to investigate this background to see how signal-like it is in the jet-substructure.. - A W-boson that is emitted close to a high-pt quark can produce the appropriate signal-like signature when the W decays to a charged-lepton and neutrino. Again, it would be appropriate to investigate this background to see how signal-like it is in the jet-substructure.

(2) The investigated multijet background is produced using Pythia8, implying it is generated as a 2->2 scatter and showered. However, the signal is produced with up to 4 particles in the matrix-element using Madgraph. Given the DM can then instigate a SM shower, it means the signal effectively has up to 4-jets at ME level. It is not clear to what level the difference between the signal and background is induced by the modelling differences between Madgraph+Pythia and pure Pythia.

(3) Experimentally, the MET+jets signature at the LHC is investigated by requiring that the jet and the missing transverse momentum are well separated in azimuthal angle (dphi(MET,jet)>0.4). The reason is that jets are mismeasured in calorimeters, which induces a MET signature. The analysis done here proposes to select events with dphi(MET,jet)<1.0, to understandably capture the missing transverse momentum close-by the jet. However, no discussion is given in the paper as to the experimental difficulties of doing this.

(4) I found it very difficult to understand the final states that are investigated. Section 3 defines that we produce the signal with up to 2 extra partons, does that mean we generate at ME-level the following: (i) SM SM -> DM DM, (ii) SM SM -> DM DM SM, (iii) SM SM -> DM DM SM SM? The exact generation needs to be more explicit.

(5) The theory section has a lot of discussion that eventually is not needed (the effective lagrangian) and then few details on the explicit model used. For example, there seems to be a coupling parameter lambda, which affects the cross section of the process in figure 3, that is not discussed at all. The theory section should focus on clearly explaining the model that was generated.

Report

The expectations are met, in that the research explores a new research direction and, if successful, could lead to many measurements from the ATLAS and CMS experiments.

The general criteria are mostly met, but a clearer theory section to explicitly state how the signal and background topologies are generated would be very useful (general criterion 3). In addition, the text could be made a little clearer in quite a few places (general criterion 1).

It is my opinion that any deficiencies can be addressed by the authors and the paper can satisfy all criteria to be published in SciPost Physics.

Requested changes

1) Add investigation of more backgrounds: - Z+jets (Z->tautau). - W+jets (W->lnu, l=e,mu,tau)

(2) Investigate the impact of using Madgraph+Pythia8 for the multijet background, with up to 4 partons in the final state). Ideally, could just change the background model to use Madgraph+Pythia directly.

(3) Add a discussion about experimental considerations, given that current MET+jets searches impose a cut on dphi(MET,jet)>0.4 to remove the impact of mis-measured jets. The goal is to explain how experimenters can avoid being affected by mismeasured jets if searching for semi-visible jets at the LHC.

(4) Improve the theory section: - add a discussion to explain exactly what is generated for the signal, and include Feynman diagrams as appropriate. Similarly, need to fully explain the background process physics, rather than just referring to "HardQCD" process. - improve the discussion of the model in the theory section, including a full description of all parameters and focussing on aspects that affect this particular analysis (rather than general considerations of effective lagrangians)

(5) Some aesthetics: - why not use a subscript "d" for all dark mesons. Makes it easier to read. - can the legend labelling be improved? i.e. in fig 4 Signal, r_inv=0.5 is far more meaningful than Sig50. - C2 and D2 seem to be identical. Needs fixing. Why do you need to discuss D2 at all? there are no plots of D2 in the paper.

  • validity: ok
  • significance: good
  • originality: good
  • clarity: ok
  • formatting: reasonable
  • grammar: reasonable

Author:  Deepak Kar  on 2021-03-14  [id 1307]

(in reply to Report 3 on 2021-02-19)
Category:
answer to question

Dear referee,

We thank you for your review, and apologise for the delay in responding. One of the authors got busy in start of the term activities, other was moving countries to start her MCNet short-term student ship.

We completely agree with your assessment, this is meant to be a first exploratory study, far from the last word on this exciting area. Your suggestions definitely improve the quality of the paper. Please find our responses inline for your suggested changes:

( 1) Add investigation of more backgrounds: - Z+jets (Z->tautau). - W+jets (W->lnu, l=e,mu,tau)

We have expanded the text by adding a paragraph discussing the subdominant backgrounds:

Apart from the multijet process, which is the dominant background, $W/Z$ + jets processes can contribute to the background. However, the processes with one or more leptons can be almost completely rejected by vetoing events with leptons. The $W \rightarrow \tau (\textrm{had})\nu$ and $Z \rightarrow \nu\nu$ processes result in irreducible backgrounds, but since the large radius jets will still come from quarks or gluons, the considerations for multijet events still hold.

(2) Investigate the impact of using Madgraph+Pythia8 for the multijet background, with up to 4 partons in the final state). Ideally, could just change the background model to use Madgraph+Pythia directly.

We have generated events with Madgaph (upto 4 partons in fina state, as suggested) +Pythia8, using CKKWL matching, to check if conclusions change, which do not. We attach the plots here for your consideration, and in text we added a line.

Please see the attachment.

(3) Add a discussion about experimental considerations, given that current MET+jets searches impose a >cut on dphi(MET,jet)>0.4 to remove the impact of mis-measured jets. The goal is to explain how >experimenters can avoid being affected by mismeasured jets if searching for semi-visible jets at the LHC.

We added:

It must be noted though, that while multijet events at particle level mostly have low values of missing transverse momentum, at detector level, due to mis-measurement of jet energy and direction, a large fraction of events acquire large values of missing transverse momentum. This is essentially an irreducible background. Another possible origin missing transverse momentum close to jets arise the jet area includes dead calorimeter cells. In experimental searches, this is typically accounted for by removing jets when the angular separation of jets and missing transverse momentum direction is less than $\Delta \phi < 0.4$. For a signature like this, that requirement is clearly unusable, but events where jet areas include dead calorimeter cells can be discarded, and in ATLAS this is seen to be a small fraction of events~\cite{Aad:2020aze}.

(4) Improve the theory section: - add a discussion to explain exactly what is generated for the signal, and include Feynman diagrams as >appropriate. Similarly, need to fully explain the background process physics, rather than just referring to "HardQCD" >process. - improve the discussion of the model in the theory section, including a full description of all >parameters and focussing on aspects that affect this particular analysis (rather than general >considerations of effective lagrangians)

We expanded the text (as best as us experimentalists can do!), and added Feynman diagrams as requested.

(5) Some aesthetics: - why not use a subscript "d" for all dark mesons. Makes it easier to read.

Done

  • can the legend labelling be improved? i.e. in fig 4 Signal, r_inv=0.5 is far more meaningful than Sig50.

Done

  • C2 and D2 seem to be identical. Needs fixing. Why do you need to discuss D2 at all? there are no plots >of D2 in the paper.

We think it adds some information to say D2 was seen to be similar, so kept that.

The revised draft has been submitted to arXiv, it should appear Tuesday (16th March) morning.

Best regards, Sukanya and Deepak

Attachment:

ObsComp.pdf

---

## Round 4 · Referee Report · Tilman Plehn (Referee 2) · 2021-3-16

Report

Thanks for considering my comments. Let's roll!
  • validity: -
  • significance: -
  • originality: -
  • clarity: -
  • formatting: -
  • grammar: -

Author:  Deepak Kar  on 2021-04-14  [id 1358]

(in reply to Report 1 by Tilman Plehn on 2021-03-16)

Thanks Tilman for your support!

---

## Round 4 · Referee Report · Jonathan Butterworth (Referee 1) · 2021-4-1

Report

Thanks for the response and the changes made. I have one remaining request - I agree with your estimate/statement on pile up, but I since this is often considered under "detector effects" I think you should definitely make it clear in the paper that you did not study it (and also say, if you like, that because you used trimmed jets it should not be a big effect).

With that addition (or even without it, if you really don't want to include it!) I would be happy to see this published.

Requested changes

1 include statement on pile up

  • validity: -
  • significance: -
  • originality: -
  • clarity: -
  • formatting: -
  • grammar: -

Author:  Deepak Kar  on 2021-04-14  [id 1357]

(in reply to Report 2 by Jonathan Butterworth on 2021-04-01)

Thanks Jon, we added a line:

The effect of pile-up was not considered as well, but use of groomed jets (trimming was used here) should mitigate the effect of it to a large extent.

Jonathan Butterworth  on 2021-04-14  [id 1360]

(in reply to Deepak Kar on 2021-04-14 [id 1357])

Thanks. All good to be published from my side.

---

## Round 4 · Referee Report · Anonymous (Referee 3) · 2021-4-12

Report

Thanks for considering my comments and adding some clarity to the paper in the different sections.

My main comment was on the V+jets backgrounds. It would have been nice to see simulations of the processes because it is not clear, for example, that the lepton identification in a densely populated jet would be good enough to reject W+jets entirely . That said, it is fine that these backgrounds are acknowledged in the paragraph you have added to that section.

Perhaps the wording could be slightly softened by (i) removing the 'almost completely' before 'rejected' and (ii) stating that the study of these backgrounds is left to a future publication.
  • validity: -
  • significance: -
  • originality: -
  • clarity: -
  • formatting: -
  • grammar: -

Author:  Deepak Kar  on 2021-04-14  [id 1359]

(in reply to Report 3 on 2021-04-12)

Dear referee,

Thanks for your report.

We have made the following change based on your suggestion:

However, the processes with one or more leptons can be almost completely rejected by vetoing events with leptons. The $W \rightarrow \tau_{\textrm{had}}\nu$ and $Z \rightarrow \nu \nu$ processes result in irreducible backgrounds, but since the large radius jets will still come from quarks or gluons, the considerations for multijet events still hold. -
->
However, the processes with one or more leptons can be rejected by vetoing events with leptons. The $W \rightarrow \tau_{\textrm{had}}\nu$ and $Z \rightarrow \nu \nu$ processes result in irreducible backgrounds, but since the large radius jets will still come from quarks or gluons, the considerations for multijet events still hold. A detailed study of these backgrounds is left for future work.

Best regards,
The authors

---

## Round 4 · Author Response

Dear editor,

We thank the referees for their reviews, which have certainly improved the quality of the paper. We believe we addressed all the important points to the best of our abilities. As noted by the referees, we view this as a first exploratory study on the subject, with followups planned/possible in many directions.

Sincerely,
Sukanya and Deepak

---

## Round 4 · List of Changes

The major changes are listed:

  • The theory section has been expanded, adding the relevant Lagrangian and explaining the parameters of the model.

  • The description of signal and background event generation has been expanded, and cross-checks mentioned.

*An angular event display is added for some representative events.

  • Added text about possible experimental backgrounds.

  • Added or clarified interpretation of the behaviour seen in plots.

  • Added more text about possible detector effects.

  • Figure legends and symbols cleaned up/better explained.

  • Added more relevant citations

---

## Editorial Decision

published